# Designing a 1550 nm Pulsed Semiconductor Laser-Emission Module Based on a Multiquantum-Well Equivalent Circuit Model

Li Li [1,2], Lin Li [1,*], Huiwu Xu [3], Lihua Yan [3], Gang Li [2], Dapeng Wang [3], Jiaju Ying [2] and Fuyu Huang [2]

1    School of Optoelectronics, Beijing Institute of Technology, No. 5 South Zhongguancun Street, Beijing 100081, China
2    Department of Opto-Electronics Engineering, Shijiazhuang Campus, Army Engineering University of PLA, No. 97, West Heping Road, Shijiazhuang 050000, China
3    The 13th Research Institute of China Electronics Technology Group Corporation, No. 113, Hezuo Road, Shijiazhuang 050051, China
*    Correspondence: bit421@bit.edu.cn

**Abstract:** The demand for eye-safe 1550 nm pulsed semiconductor laser-emission modules is increasing in the field of active laser detection, owing to their long range and high precision. The high power and narrow pulse of these modules can significantly improve the distance and accuracy of active-laser detection. Here, we propose an equivalent circuit model of a multiquantum-well laser based on the structure of a laser device. We developed a design method for 1550 nm pulsed semiconductor laser-emission modules according to the equivalent circuit model of an InGaAlAs laser. In this method, the module design was divided into laser chip and laser-driver levels for optimization and simulation. At the chip level, a high-output power laser chip with optimal cavity length and optical facet coating coefficients was obtained. At the laser-driver level, the model was applied to a drive circuit to provide direct narrow optical pulses. Finally, a laser-emission module was fabricated based on the optimal design results. In addition to the power-current features of the actual laser, the critical voltage of the emission module and laser pulses were tested. By comparing the test and simulation results, the effectiveness of the proposed method was confirmed.

**Keywords:** laser optics; 1550 nm multiquantum-well laser; pulsed laser emission; equivalent circuit model; narrow pulses; high power

## 1. Introduction

Pulsed semiconductor laser-emission modules, which provide narrow laser pulses with high optical power peaks, have been widely used in automobiles, Lidar, adaptive cruise control, etc. In the military, these modules are mainly applied to ranging, short-range illumination, enemy identification, and weapon simulation for training [1–4]. Compared to conventional high-power semiconductor lasers (working wavelengths of 800–1420 nm), the 1550-nm-wavelength laser has several advantages, namely less damage to human eyes [5], strong ability to penetrate smoke, low attenuation in the air, less solar background noise, and expensive detection. As its detection involves a high cost, it is advantageous in military photoelectric countermeasures. Therefore, 1550 nm pulsed semiconductor laser-emission modules attract considerable attention from the military of various countries and excite the interest of researchers investigating civil–military dual-purpose technologies [6].

The research and development of a 1550 nm pulsed semiconductor laser-emission module need to focus on the following aspects: (i) optimizing the design of the laser chip for the high-power pulsed operation mode [7], and (ii) designing the semiconductor laser's drive power to match to the laser chip [8,9].

Conventionally, the 1550 nm semiconductor laser chip adopts the InGaAsP/InP material system. When the working temperature is high, this laser device is significantly

impacted and fails to reach a high level of efficiency or high-output power. This is attributed to the low-electron and hole-confinement ability at high temperatures. The InGaAlAs/InP material system has better temperature properties than InGaAsP/InP [10]. At high temperatures, the InGaAlAs quantum well confines electrons and holes more effectively than the InGaAsP quantum well [11]. Thus, InGaAlAs/InP can be utilized to make semiconductor quantum-well lasers that perform well at high temperatures [12]. To further enhance the output power and efficiency of the laser, an asymmetric separate confinement heterostructure (SCH) can be adopted to reduce the internal optical losses in the quantum well laser.

In most studies conducted on semiconductor laser drive power, the laser is replaced with equivalent loads, such as power diodes. However, power diodes convert all the electrical energy into thermal energy and fail to simulate the electro-optical conversion features of semiconductor lasers [13]. Therefore, a semiconductor laser model can be utilized to simulate the laser-pulse curves.

For GaAs-based lasers with wavelengths in the range of 850–1060 nm, simulation models are used to optimize the heterostructure parameters, cavity length, and reflection coefficients for high-power pulsed lasers. Wang et al. conducted a systematic study to determine the sources that limit the peak output power of broad-area single-emitter diode lasers under high current and pulse-pumped operation conditions [14]. Considering the importance of carrier transport, Soboleva et al. were the first to examine the effects of drift velocity saturation at an ultrahigh drive current densities using the classical drift–diffusion transport and energy-balance models [15]. In addition, two-dimensional electro-optic models have been employed for evaluating the GaAs-based lasers [16,17]. In this study, we adopted actual device parameters and the rate equation to establish an equivalent circuit model of a 1550 nm InGaAlAs/InP multiquantum-well semiconductor laser. The primary aim of this study was to analyze the effect of cavity length and optical facet coating coefficients on the output optical-pulse features and to optimize the laser structure for peak-pulse operations. The developed laser model was used as an equivalent load, reflecting the electro-optical features of the laser, and connected to the drive circuit for optimizing the drive power design. Thus, we obtained an optical pulse with a width of less than 10 ns, a rise time of approximately 4 ns, and a maximum output power of 20 W. The laser-emission module can generate laser outputs with high powers and narrow pulses. The experimental power–current (PI) data and test waveforms of the module were consistent with the simulation results. These results validate the effectiveness of the developed model and our design method.

## 2. Structure and Equivalent Circuit Model of the 1550 nm InGaAlAs/InP Laser

A 1550 nm single-junction semiconductor laser with an InGaAlAs/InP material system was employed in this study. The structure of the laser is illustrated in Figure 1. The active region of the laser has a strain-compensated multiquantum-well structure containing InGaAlAs wells and barrier layers. The optical confinement factor and gain coefficient of a multiquantum-well structure are higher than those of a single-quantum-well structure. The active region is located at the centre of the optical confinement layer of the graded index separate confinement heterostructure (GRIN-SCH). Due to the asymmetric SCH design, the eigenmode distribution of the laser is biased toward the n-type cladding, which reduces the overlap between the laser eigenmode and the p-type-doped cladding. This feature helps in reducing the internal optical loss, thus enhancing the output power of the semiconductor laser [18].

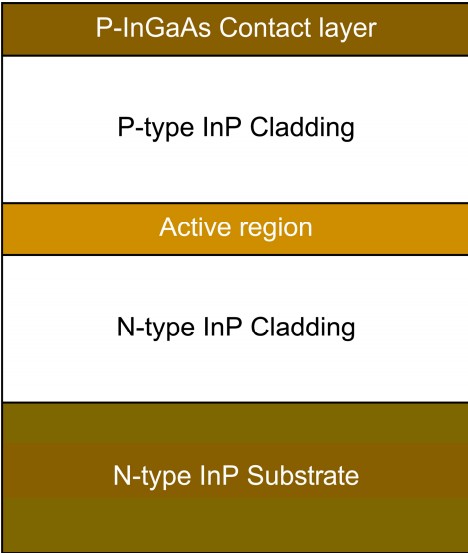

**Figure 1.** Laser structure.

The laser-circuit model was built based on the carrier-transport behaviour, which can be classified into spatial transport and state transport in an SCH quantum-well laser. Spatial transport refers to the drift and diffusion of the carriers in the SCH region, while state transport refers to the capture and escape of the carriers in the quantum wells. Considering the above transport behaviour, the carriers were classified into three states: the three-dimensional carriers in the SCH region, two-dimensional carriers in the confined state of the quantum wells, and quasi-two-dimensional carriers in the quantum well with higher energy than the confined state, and whose motion falls between those of two and three dimensions [19]. The carriers are assumed to pass through the potential barrier without undergoing recombination.

Let $n$, $s$, $R$, $V$, $\tau$, $G(n,s)$, and $L$ denote the carrier density, photon density, carrier recombination, volume of each region, effective spatial transmission time, optical gain, and thickness of each region, respectively. Further, subscripts $S1$, $b$, $w$, and $S2$ denote the P-side SCH region, quasi-two-dimensional region, two-dimensional region, and N-side SCH region, respectively.

Then, the carrier density in the P-side SCH region can be expressed as follows:

$$\frac{dn_{S1}}{dt} = \frac{I_j}{qV_{S1}} - R_{S1} - \frac{I_{D,1}}{qV_{S1}}, \tag{1}$$

where $I_j$ is the injection current, $q$ is the electron charge, $I_{D,1} = \frac{q(V_{S1}n_{S1} - V_w n_{b1})}{\tau_{S1}}$ is the drift-diffusion current in the P-side SCH region with $\tau_{S1} = \frac{L_{S1}^2}{2D_a}$, $\tau_{S1}$ is the effective spatial transport time, and $D_a$ is the bipolar diffusion coefficient.

In the j-th well, the carrier density in the quasi-two-dimensional region can be expressed as follows:

$$\frac{dn_{b,j}}{dt} = \frac{I_{D,j}}{qV_w} - \frac{I_{D,j+1}}{qV_w} - R_{b,j} - \left( \frac{n_{b,j}}{\tau_c} - \frac{n_{w,j}}{\tau_e} \right). \tag{2}$$

When $j \neq 1$, $I_{D,j} = \frac{q(V_w n_{b,j-1} - V_w n_{b,j})}{\tau_b}$ represents the diffusion current across the potential barrier, $\tau_c$ and $\tau_e$ are the capture time and escape time of the carrier, respectively.

In the j-th well, the carrier density in the two-dimensional region can be expressed as follows:

$$\frac{dn_{w,j}}{dt} = \left( \frac{n_{b,j}}{\tau_c} - \frac{n_{w,j}}{\tau_e} \right) - R_{w,j} - v_g G(n_{w,j}, s)s, \tag{3}$$

where $v_g$ is the group velocity.

In the last well, the carrier density of the combined N-side SCH region in the quasi-two-dimensional region can be expressed as follows:

$$\frac{dn_{S2}}{dt} = \frac{I_{D,m}}{qV_{S2}} - \frac{I_{Dout}}{qV_{S2}} - R_{S2} - \left(\frac{V_w}{V_{S2}} \cdot \frac{n_{S2}}{\tau_c} - \frac{V_w}{V_{S2}} \cdot \frac{n_{w,m}}{\tau_e}\right), \tag{4}$$

where $m$ is the number of wells; $I_{D,m} = \frac{q(V_w n_{b,m-1} - V_w n_{S2})}{\tau_b}$, with $\tau_b = \frac{L_b^2}{2D_a}$; and $I_{Dout} = \frac{qV_{S2} n_{S2}}{\tau_{S2}}$, with $\tau_{S2} = \frac{L_{S2}^2}{2D_a}$.

The photon density can be expressed as follows:

$$\frac{ds}{dt} = \sum_{j=1}^{m} \Gamma_j v_g G(n_{w,j}, s)s + \sum_{j=1}^{m} \Gamma_j \beta_{sp,j} B_{w,j} n_{w,j}^2 - \frac{s}{\tau_{ph}}, \tag{5}$$

where $\Gamma$ is the optical confinement factor, $\beta_{sp}$ is the spontaneous emission factor, and $\tau_{ph}$ is the photon lifetime. $\tau_{ph}$ can be expressed as follows:

$$\tau_{ph} = \frac{\overline{n}_g}{c\left(\alpha_{\text{int}} - \frac{\ln(R_R R_L)}{2L}\right)}, \tag{6}$$

where $\overline{n}_g$ is the group refractive index; $\alpha_{\text{int}}$ is the cavity loss; $L$ is the cavity length; and $R_R$ and $R_L$ are the reflectance values of the right and left optical facets, respectively. The gain can be expressed as follows:

$$G(n,s) = \frac{G_0}{1 + \varepsilon s}\left(1 + \ln\frac{n}{N_0}\right), \tag{7}$$

where $G_0$ is the gain constant, $N_0$ is the transparent carrier density, and $\varepsilon$ is the gain suppression factor.

Numerically, the carrier recombination can be expressed as:

$$R = An + Bn^2 + Cn^3, \tag{8}$$

where $A$ is the Shockley–Read–Hall-type nonradiative recombination coefficient, $B$ is the radiative recombination coefficient, and $C$ is the Auger recombination coefficient.

The above carrier density equations for each region of the laser were mathematically transformed to establish certain connections with the circuit structure. All the variables and known parameters that need to be solved were transformed into circuit components, forming an equivalent circuit model for a specific laser.

Suppose $\overline{V}_{S1} = qn_{S1}$, $\overline{V}_{S2} = qn_{S2}$, $\overline{V}_{b,j} = qn_{b,j}$, $\overline{V}_{w,j} = qn_{w,j}$, and $\overline{V}_s = qs$. By equating these parameters to the node voltages in the circuit, Equations (1)–(5) can be transformed into Equations (9)–(13) as follows:

$$I_j = V_{S1}\frac{d\overline{V}_{S1}}{dt} + I_{rS1} + I_{D,1}, \tag{9}$$

where $I_{rS1} = qV_{S1}R_{S1}$.

Considering the current flowing through the capacitor $I_C = C\frac{dV_C}{dt}$, the term $V_{S1}\frac{d\overline{V}_{S1}}{dt}$ in Equation (9) can be equivalent to the current flowing through the capacitor. The capacitor value is $V_{S1}$, and the voltage at both ends of the capacitor is $\overline{V}_{S1}$. Similarly, the time-varying terms in Equations (10)–(13) were related to the current of the capacitor node as follows

$$I_{D,j} + I_{e,j} = V_w\frac{d\overline{V}_{b,j}}{dt} + I_{rb,j} + I_{c,j} + I_{D,j+1}, \tag{10}$$

where $I_{rb,j} = qV_w R_{b,j}$, $I_{c,j} = V_w \frac{\overline{V}_{b,j}}{\tau_c}$, and $I_{e,j} = V_w \frac{\overline{V}_{w,j}}{\tau_e}$.

$$I_{c,j} = V_w \frac{d\overline{V}_{w,j}}{dt} + I_{rw,j} + I_{e,j} + I_{st,j}, \tag{11}$$

where $I_{rw,j} = qV_w R_{w,j}$ and $I_{st,j} = V_w v_g G(n_{w,j}, s)\overline{V}_s$.

According to Equation (8), the compound current $I_{rw,j}$ can be divided into a no-radiative compound current ($I_{nrw,j}$) and a spontaneous radiative compound current ($I_{sp,j}$). In this case, $I_{nrw,j} = qV_w \left(An_{w,j} + Cn_{w,j}^3\right)$ and $I_{sp,j} = qV_w Bn_{w,j}^2$.

$$I_{D,m} + I_{e,m} = V_{S2} \frac{d\overline{V}_{S2}}{dt} + I_{Dout} + I_{rS2} + I_{c,m}, \tag{12}$$

where $I_{e,m} = V_w \frac{\overline{V}_{w,m}}{\tau_e}$, $I_{rS2} = qV_{S2}R_{S2}$, and $I_{c,m} = V_w \frac{\overline{V}_{S2}}{\tau_c}$.

$$I_{st} + I_{sp} = V_w \frac{d\overline{V}_s}{dt} + \frac{\overline{V}_s}{R_{ph}}, \tag{13}$$

where $I_{st} = \sum_{j=1}^{m} \Gamma_j I_{st,j}$, $I_{sp} = \sum_{j=1}^{m} \Gamma_j qV_w \beta_{sp,j}B_{w,j}n_{w,j}^2$, and $R_{ph} = \frac{\tau_{ph}}{V_w}$.

The equivalent circuit model of the laser was obtained by expressing the equations in circuit form, as shown in Figure 2. The voltage $\overline{V}_s$ characterizes the number of photons generated in the resonant cavity. To obtain the output power, this voltage is converted to detect the optical power directly at the model output ends.

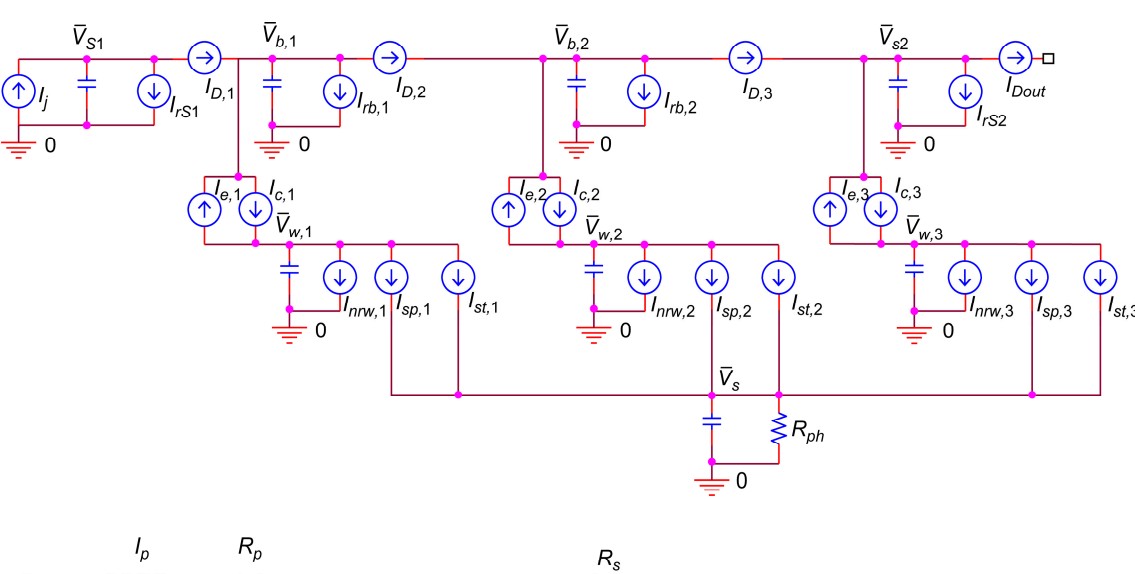

**Figure 2.** Equivalent circuit model of a multiquantum-well laser (considering 3 quantum wells as an example).

The output optical power P at the left and right optical facet of the laser is related to the photon density as follows [20]:

$$
\begin{aligned}
P_L &= \frac{hc^2 WD(R_L-1)\ln(R_L R_R)}{2q\overline{n}_g \lambda \left[1-R_L+\sqrt{R_L/R_R}(1-R_R)\right]} \overline{V}_s \\
P_R &= \frac{hc^2 WD(R_R-1)\ln(R_L R_R)}{2q\overline{n}_g \lambda \left[1-R_R+\sqrt{R_R/R_L}(1-R_L)\right]} \overline{V}_s,
\end{aligned}
\tag{14}
$$

where $\lambda$ is the wavelength in a vacuum, $D$ is the total thickness of the active and SCH regions, and $W$ is the width of the active region.

In pulsed mode, the semiconductor laser operates with a large current. The parasitic parameters are non-negligible in the start-up and steady-state processes. These parameters must be considered when modeling the laser, as shown in Figure 2. The parasitic parameters of a semiconductor laser include the resistance $R_p$, capacitance $C_p$, inductance $L_p$ generated in the package, parasitic parallel capacitance $C_s$, series resistance $R_s$, and parallel resistance $R_d$ generated inside the chip. The current $I_j$ injected into the intrinsic region can be equated with the diode equation $I_j = I_S \left[ \exp\left( \frac{V_j}{\eta V_T} \right) - 1 \right]$, where $I_S$ is the diode reverse saturation current, and $\eta$ is the ideal factor. Further, $V_T = \frac{kT}{q}$, where $k$ is the Boltzmann constant and $T$ is the absolute temperature. The corresponding values of the model parameters are summarized in Table 1.

**Table 1.** Values of the model parameters.

| Symbol | Parameters | Value | Unit |
|--------|------------|-------|------|
| $\lambda$ | Wavelength | 1550 | nm |
| $L$ | Cavity length | To be optimized | mm |
| $W$ | Width of the active region | 90 | μm |
| $L_{S1}$ | Thickness of the P-side SCH region | 1 | μm |
| $L_w$ | Thickness of well | 7 | nm |
| $L_b$ | Thickness of barrier | 10 | nm |
| $L_{S2}$ | Thickness of the N-side SCH region | 0.85 | μm |
| $m$ | The number of wells | 5 | – |
| $\beta_{sp}$ | Spontaneous emission factor | 0.0001 | – |
| $\Gamma$ | Optical confinement factor | 0.02 | – |
| $A$ | Shockley–Read–Hall recombination coefficient | $2.2 \times 10^8$ | $\text{s}^{-1}$ |
| $B$ | Spontaneous radiative recombination coefficient | $0.8 \times 10^{-10}$ | $\text{cm}^3\text{s}^{-1}$ |
| $C$ | Auger recombination coefficient | $3.6 \times 10^{-29}$ | $\text{cm}^6\text{s}^{-1}$ |
| $D_a$ | Bipolar diffusion coefficient | $7 \times 10^{-4}$ | $\text{cm}^2\text{s}^{-1}$ |
| $G_0$ | Gain constant | 6400 | $\text{m}^{-1}$ |
| $N_0$ | Transparent carrier density | $1 \times 10^{23}$ | $\text{m}^{-3}$ |
| $\varepsilon$ | Gain suppression factor | $4 \times 10^{-23}$ | – |
| $\tau_c$ | Capture time | 0.1 | ns |
| $\tau_e$ | Escape time | 10 | ns |
| $\alpha_{\text{int}}$ | Cavity loss | 6 | $\text{cm}^{-1}$ |
| $R_R\ R_L$ | Reflectances of the optical facets | 0.3 (for natural cleavage surface) | – |
| $\overline{n}_g$ | Group refractive index | 3.42 | – |

## 3. Optimized Design of Pulse Emission Module

The laser model presented in the previous section needs to be modified based on the device parameters of the actual laser (Table 1) before being used to simulate an actual laser.

### 3.1. Optimized Design for High Output Power of Laser Chip

The output power of the chip is generally increased by optimizing the quantum well structure, increasing the chip cavity length, and adjusting the coating reflectivity. The

quantum well structure must be designed within the constraints of the optical field and carriers. In this study, we designed a chip containing an InGaAlAs well layer with a compressive strength of 1.1%, an InGaAlAs barrier layer with a tensile strength of 0.6%, a InAlAs–InGaAlAs graded-index waveguide layer, InP cladding, and five quantum wells.

The PI features of the established laser model were simulated using SPICE. The results of different cavity lengths, L, are shown in Figure 3.

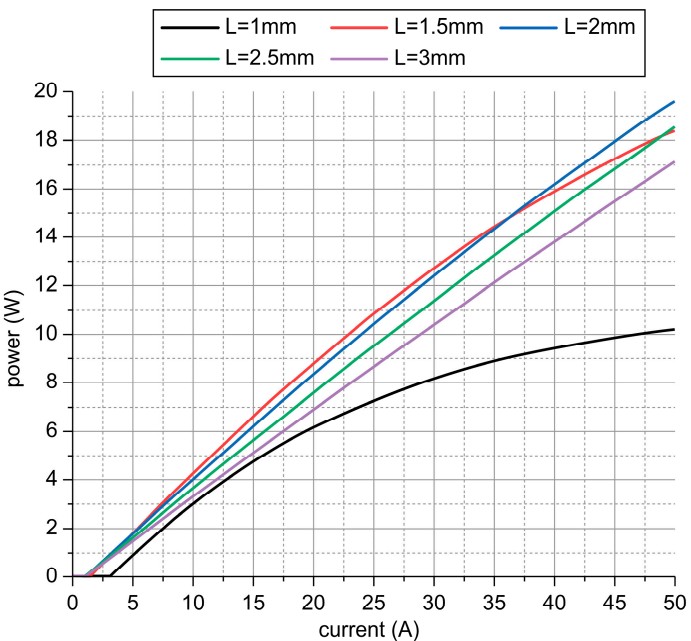

**Figure 3.** PI features simulated at different cavity lengths *L* (the reflectance values of the two optical facets is 0.3).

The simulation results showed that a longer cavity increases the chip power saturation point, such that the output power increases with the high injection current of the pulsed operation mode. Moreover, the output loss decreases with the increasing cavity length and, in turn, reduces the threshold current of the chip. However, with further elongation of the cavity, an excessively long cavity increases the cavity losses and, thus, reduces the slope efficiency. Consequently, this increases the power consumption of the chip [21]. As can be seen from Figure 3, the chip with a 2-mm-long cavity has the highest slope efficiency for this active region structure, and there is no premature saturation.

Considering the threshold current, peak working current, maximum working temperature (a short cavity has a low maximum working temperature), and packaging difficulty (a long cavity is relatively difficult to package), a laser chip with a wide aperture (90 µm) and a long cavity (2 mm) was selected to reduce the junction temperature at a given working current and power, reduce the thermal resistance, and output a large optical power [22]. When the optimized cavity is 2 mm long, the corresponding threshold current is 1 A, and the slope efficiency before the end-face coating is 0.41 W/A.

In the resonant cavity, the front optical facet is the laser exit surface. The front and rear optical facets need to have high transmittance and high reflectance, respectively. This is to ensure that most of the generated light is emitted forward and that the laser cavity has sufficient optical feedback. Notably, the reflectivity of the two optical facets affects the external differential quantum efficiency and threshold current density of the laser. If the rear optical facet is coated with multiple layers of λ/4 thick dielectric films with alternating high and low refractive indices, the reflectivity will be close to 100%. Considering the techniques employed in the real world, the rear optical facet reflectivity was set to 98%. Notably, there exists an optimal front optical facet reflectivity for the external differential quantum efficiency. The established laser model was simulated with variable front optical facet

reflectivity values, revealing the effects of front optical facet reflectivity on the steady-state features of the laser. Thus, the optimal front reflectivity was identified in this study. Figure 4 shows the PI features of the laser with different front optical facet reflectivity values.

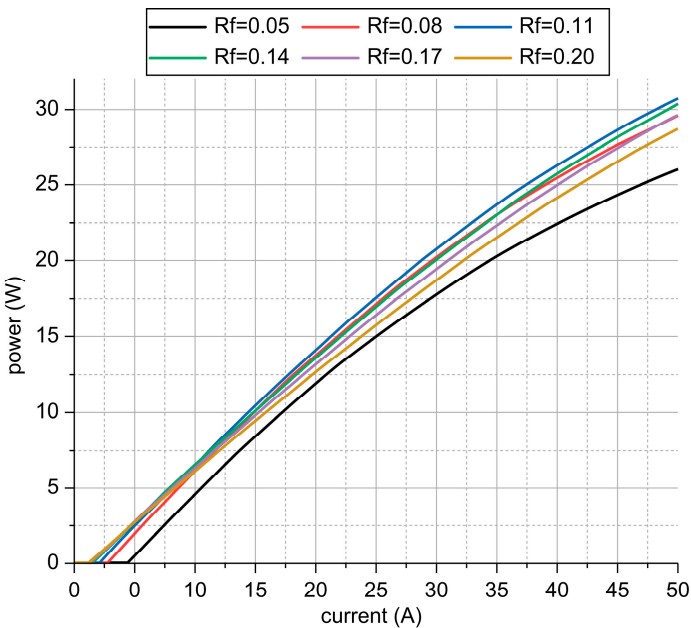

**Figure 4.** PI features simulated at different front optical facet reflectivity values (for a cavity length of 2 mm).

As shown in Figure 4, the higher the front optical facet reflectivity, the lower the threshold current, and the better the slope efficiency. When the reflectivity is increased beyond 11%, the slope efficiency declines with the increasing reflectivity.

Considering the high current driven by the laser in the pulsed state, the threshold current is not a concerning factor. In practical applications, it is more critical to focus on the high output power from the increased slope efficiency. Therefore, our design sets the optimal reflectivity of the front optical facet to 11%.

For a variable cavity length and reflectivity, the simulated PI features are the same as those reported previously. This consistency between the previously reported and simulated PI features demonstrates the effectiveness of the equivalent circuit model as a suitable effective method for device optimization.

### 3.2. Drive Power Design

Based on time-of-flight ranging, the active laser-detection system measures distances using the following principle. The power supply is driven to generate a narrow pulse of high current, which in turn drives the laser to generate light pulses to irradiate the object. The laser is partly reflected by the object and then received by the detector. The distance of the object is obtained by calculating the time difference between the emission and reception of the laser [2]. The precise ranging of the system requires the output of a narrow laser pulse between 1 and 25 ns; the peak current should ideally reach tens of amperes, resulting in a sufficiently fast-rising edge of the laser pulse.

Typically, a pulsed-laser drive power uses a semiconductor switch in series with the laser and an electrical energy source. Figure 5 depicts the principle of a capacitive discharge-type drive [23]. The driver mainly consists of a switching circuit, an energy-storage circuit, and a laser. The basic working process is as follows. In the absence of the trigger signal, the switch circuit is disconnected, and the capacitance $C_1$ is charged by the high-voltage $V_{in}$ via the charging circuit; when the trigger pulse arrives, the switch circuit is quickly connected, and the capacitance $C_1$ rapidly discharges through the laser and the switching circuit. The instantaneously generated high pulse current then drives the laser to emit light.

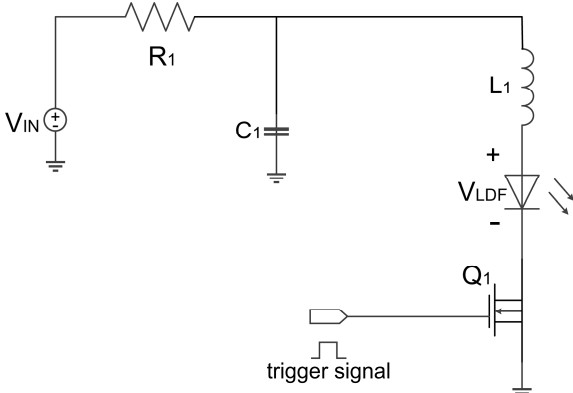

**Figure 5.** Circuit diagram of the capacitive discharge−type driver.

The capacitive discharge−type driver was designed as follows. First, the laser pulse requirements, including the pulse peak current $I_{LDpk}$, the full-width half-maximum (FWHM) pulse width $t_w$, and the laser voltage drop $V_{LDF}$ were specified. Second, the desired voltage $V_{IN}$ was determined from an estimated inductance $L_1$ using $V_{IN} = \frac{2\pi L_1}{3t_w} I_{LDpk} + V_{LDF}$, and the energy-storage capacitance $C_1$ was calculated using $C_1 = L_1 \left( \frac{I_{LDpk}}{V_{IN} - V_{LDF}} \right)^2$. The charging resistor $R_1$ was derived using $R_1 = \frac{\tau_{chrg}}{C_1}$ (generally, $R_1$ does not need to be determined precisely and it is sufficient to choose a large enough capacitor charging time constant, $\tau_{chrg}$). Finally, the appropriate switching device, $Q_1$, was selected according to $I_{LDpk}$ and $V_{IN}$.

The laser-pulse peak current $I_{LDpk}$ of 30 A can be determined based on the characteristic parameters of the actual device and its power requirements. The FWHM pulse width, $t_w$, was determined to be 7 ns. Under the estimated inductance of 1 nH, $C_1 = 10$ nF and $V_{IN} = 20$ V were identified. The switching device is the key to driving the power supply. Notably, the performance of the drive power supply is closely related to the switching time and driving capability of this switching device, directly affecting the rise time and peak power of the output laser pulse. To drive the pulsed semiconductor laser, our system used a GaN field effect transistor (FET), a high-speed switching device, to generate narrow high-power current pulses. This switching device was selected due to its high switching speed, ability to generate a large current, and ease of use. Furthermore, the driver can control the amplitude and width of the output pulse easily by replacing the energy storage capacitor and load resistor with different parameters. Figure 6 depicts the principle of the pulse drive circuit using GaN FET, which is the fundamental circuit of the drive power in the pulsed semiconductor laser.

In Figure 6, EPC2016C is an GaN FET switching device driven by the integrated circuit chip LMG1020; MQWLD is the equivalent circuit model of the laser presented in the previous section, which is added to the driving circuit as a payload using the key parameters of an actual device to simulate the electro−optical conversion features of the laser; V1 is the 20 V input high voltage; R1 is the 300 Ω charging resistor; C1 is the 10 nF energy storage capacitor; L2, R3, and C2 are the package parasitic parameters; and L1 and R2 are the loop-stray inductance and loop-stray resistance, respectively.

Figure 7 shows the simulation results for C1 = 10 nF and $V_{IN} = 20$ V. The peak current reached $I_{LDpk} = 29.2$ A, with $t_w = 7.1$ ns. The simulation results match the expected outcomes of the design calculations.

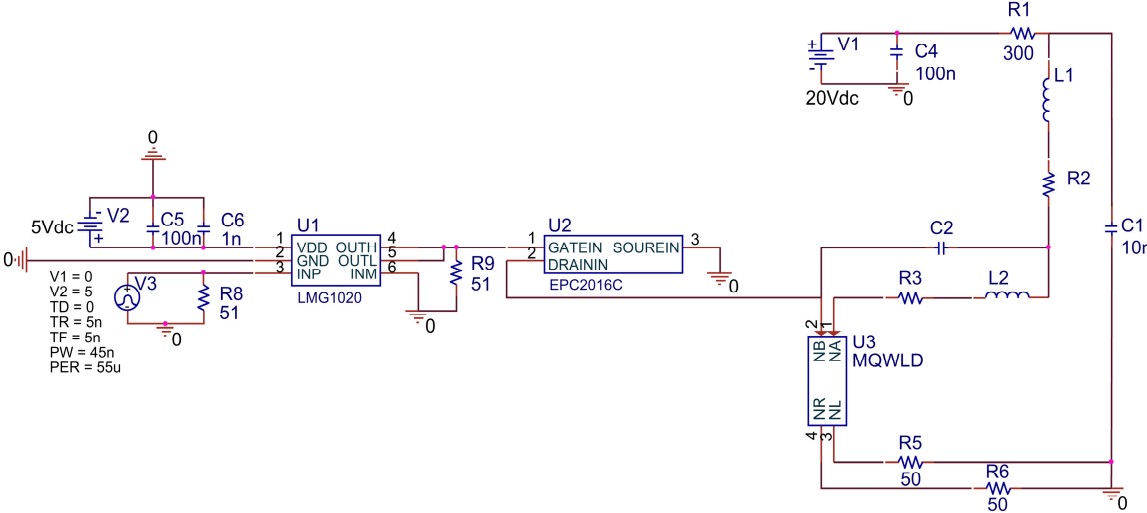

**Figure 6.** Fundamental circuit of the driver's power supply based on GaN FET.

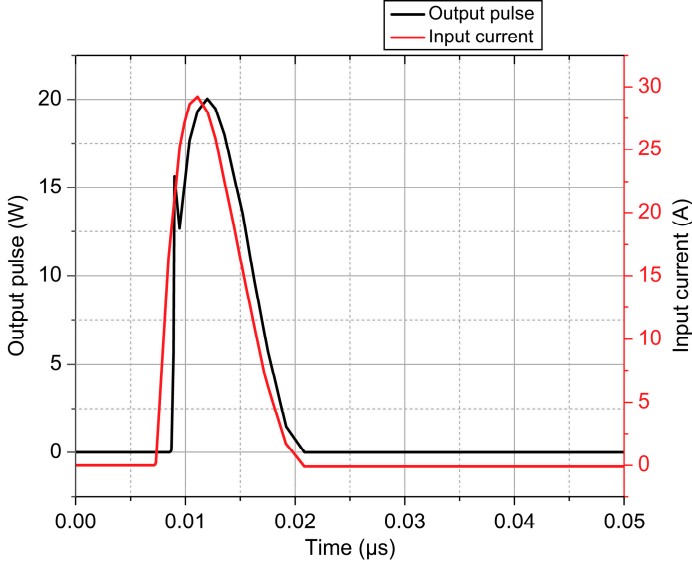

**Figure 7.** Laser-input current and output pulse waveforms.

The inductance and resistance values of the circuit have a significant effect on the performance of the pulsed circuit. Previous studies generally replaced actual semiconductor lasers with power diodes and analyzed the pulse current to indirectly infer the effect of the parameters in the circuit on the emitted laser pulses. This traditional approach cannot obtain the details of the optical pulses. Therefore, in this study, the equivalent circuit model of the actual laser was employed so that the effect of the circuit device parameters on the laser pulse can be directly analyzed, resulting in the specific data of the optical pulses.

Figure 8 shows the simulated waveforms of the laser-input current and output light pulse when the package parasitic inductor L2 was set as 0.5 nH, 1 nH, and 1.5 nH, and the other parameters of the circuit were maintained as constant. As the value of L2 increased, the pulse current and output light-pulse amplitude decreased, while the pulse widened, and the rising time of the pulse increased. The same analysis and conclusions are applicable to the loop stray inductance.

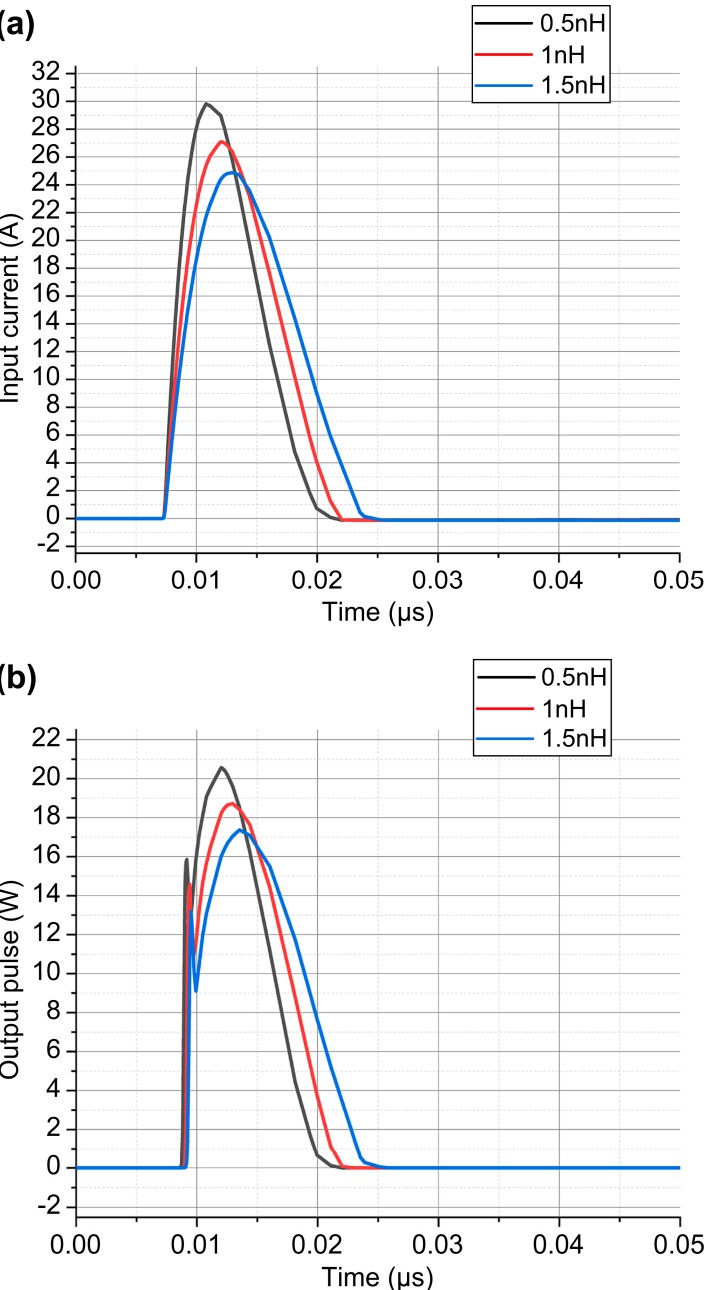

**Figure 8.** (**a**) Laser−input current and (**b**) output pulse waveforms for different package parasitic inductances.

Figure 9 shows the simulated waveforms of the laser−input current and output optical pulse when the package parasitic resistor R3 was set as 0.2 Ω, 0.6 Ω, and 1 Ω, and the other parameters of the circuit remained constant. As the value of R3 increases, both the pulse current and output light-pulse amplitude decrease significantly, while as the falling time of the pulse increases, the pulse widens. The same analysis and conclusions are applicable to the loop stray inductance.

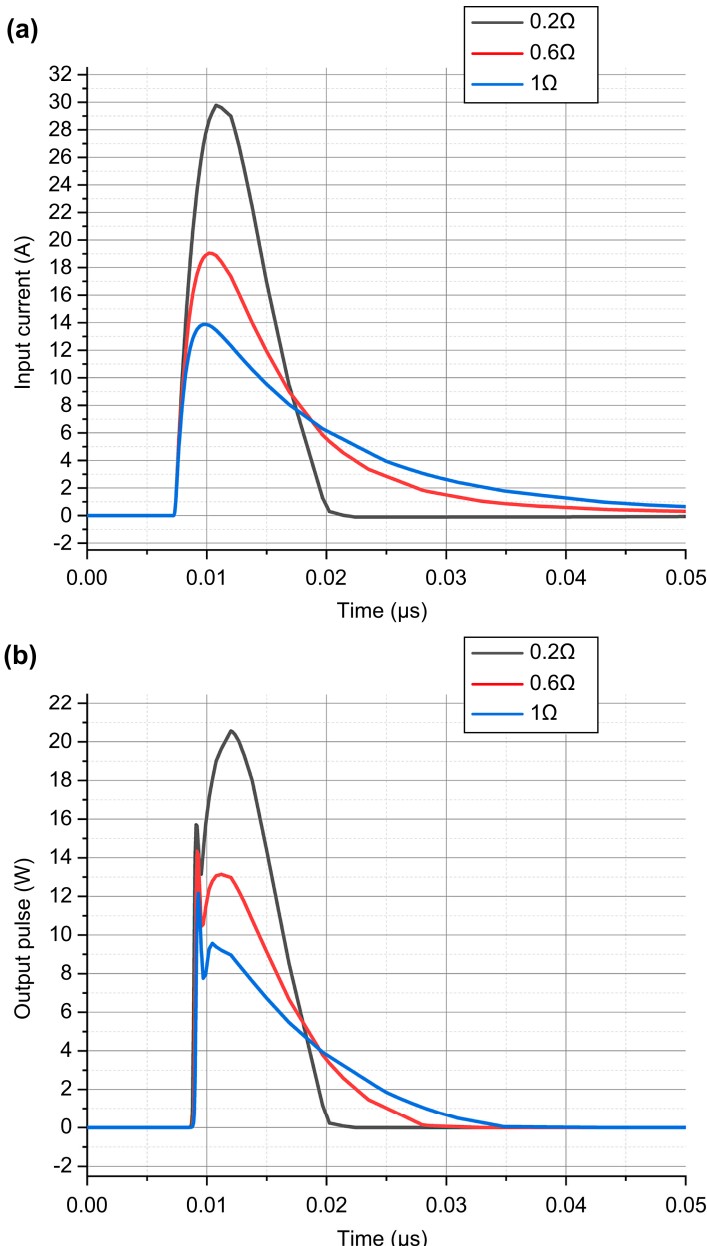

**Figure 9.** (**a**) Laser-input current and (**b**) output pulse waveforms for different package parasitic resistances.

The simulation results demonstrated that the presence of the package parasitic parameters of the laser and the stray parameters of the discharge circuit severely affected the rising/falling edge speed of the laser pulse and the peak laser power. In practice, the chip is encapsulated in the laser case, the parasitic inductance depends on the package form of the laser, and the stray inductance is caused by the layout, wiring, and via holes in the discharge circuit. Therefore, the following must be considered to control the influence of the parasitic parameters of the laser and the stray parameters of the driver circuit over the laser pulse. First, the switching circuit should use high-voltage fast switching tubes, such as the EPC2016C GaN FET in our system, which features a fast switching speed in the chip-level package. Second, the laser should be packaged with minimal inductance, such as the chip-on-board package in our system. Finally, the layout wiring of the driver circuit board should be designed with minimal total inductance.

## 4. Experimental Results

The 1550 nm InGaAlAs/InP laser with five quantum wells described in the previous section was tested, and the corresponding test results were compared with those obtained by simulations.

The PI test was performed under a pulse test condition, and the specific equipment and main parameters used for the test are listed in Table 2.

**Table 2.** PI test equipment and main parameters.

| Equipment | Main Parameters |
|---|---|
| DC power supply | 0–60 V, 3 A |
| Function Generator | Double channel, bandwidth: 200 MHz |
| Digital oscilloscope | 1 GHz, 5 GS/s |
| Peak power meter | Bandwidth: 600 MHz, Sensitivity: 0.9–1.7 μm |

A DC power supply was used to power the pulse driver, and a function generator supplied pulse signals to the pulse driver. Subsequently, the pulse driver was used to drive the chip to emit light. Further, a peak power-meter probe was used to test and detect the light pulses emitted by the chip. A digital oscilloscope was connected to the pulse driver to test the peak current. The peak-current test circuit is shown in Figure 10. To measure the current, a resistive-current shunt, formed using the five resistors R3, R4, R5, R6, and R7, was used. The measured current and optical power pulses are shown in Figure 11.

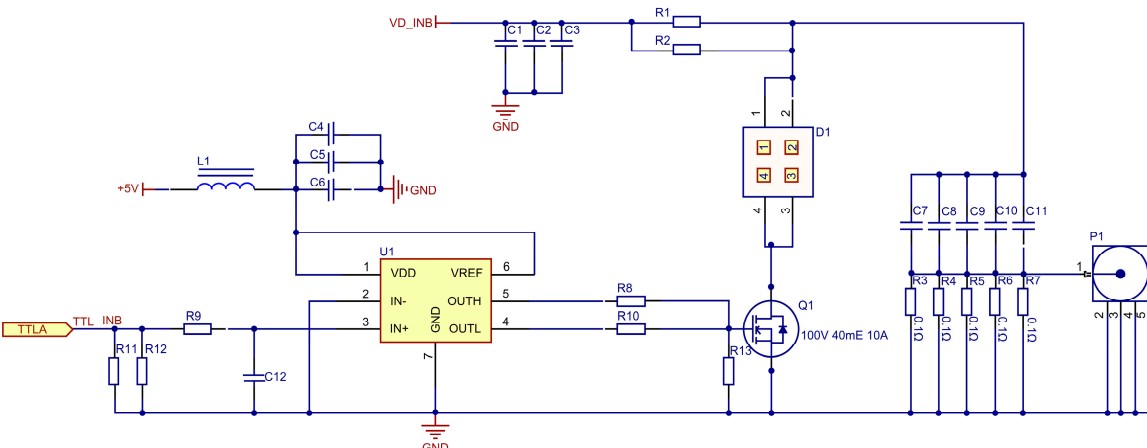

**Figure 10.** Peak current test circuit.

Figure 12 shows the measured (points) and simulated (lines) PI characteristics of the devices tested at 100 ns and 10 kHz. The measured and simulated threshold currents are similar (~2 A). Further, the laser slope efficiency reaches 0.7 W/A. A comparison of the curves reveals that the simulated one exhibits power saturation, which can be attributed to the gain compression term $(1 + \varepsilon s)^{-1}$, which is widely used as an input parameter in semiconductor laser-rate equations. Typically, gain compression is determined from small signal-intensity modulation responses of a laser, and the main associated mechanism is carrier self heating. A Fabry–Perot laser operating in the pulse mode has sufficient time to attain the equilibrium state; however, it does not heat up, and thus, the gain compression is negligible [14,16]. These results demonstrate that an efficient epitaxial material and structural design can prevent early saturation of the device.

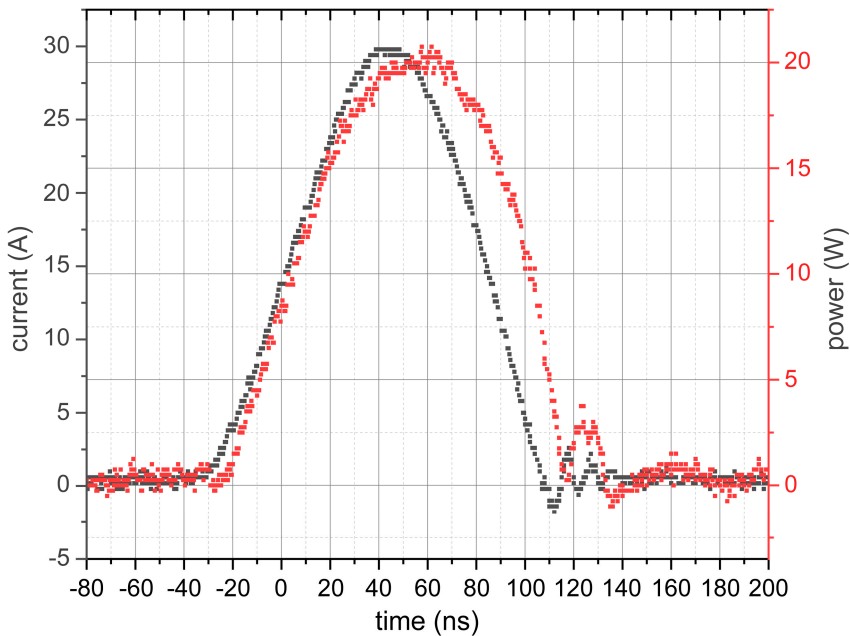

**Figure 11.** Measured current and optical power pulses.

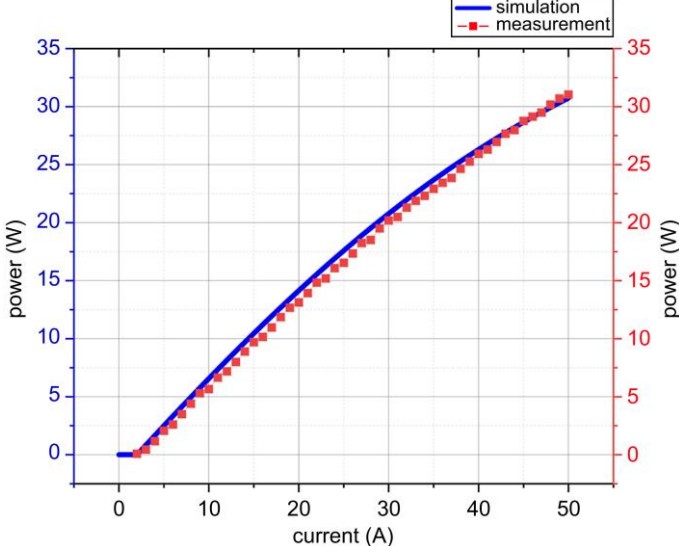

**Figure 12.** Comparison of PI features.

Based on the driver power design and the simulation analysis presented in the previous section, the actual hardware of the driver circuit board was fabricated, and the laser was mounted to form a 1550 nm pulsed semiconductor laser-emission module. This module was used to measure several major voltage waveforms in the circuit and to test the output laser pulses.

The photodetector D1 used for testing was G12180-003A with 600 MHz bandwidth, and a photodetector connection circuit is shown in Figure 13. The oscilloscope was connected to the P1 port to collect the output pulse.

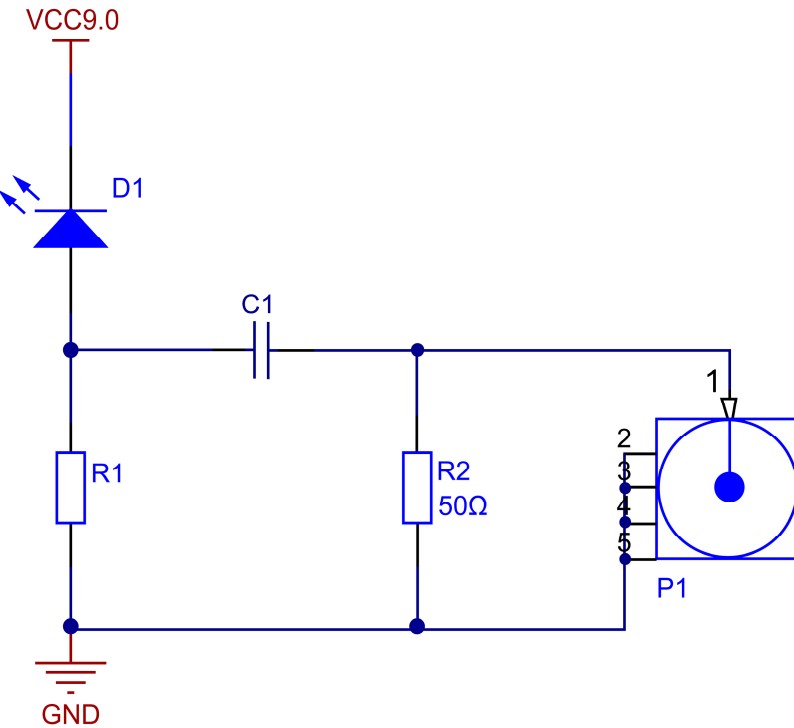

**Figure 13.** Photodetector connection circuit.

The voltages were measured at the following nodes: the switch gate, switch drain, and discharge capacitor. The predicted parasitic parameters of the circuit were adjusted based on the simulated and measured voltage waveforms. In other words, SPICE was used to fit these measured voltage waveforms to extract the inductance and resistance values, and subsequently, the simulated and experimental waveforms were compared (Figure 14) when R2 + R3 = 0.4 Ω and L1 + L2 = 1 nH.

The obtained circuit parameters reveal that the simulated laser-output pulses are in good agreement with the measured ones as shown in Figure 15. The maximum measured and simulated output-power and optical-pulse width are approximately 20 W and 10 ns (simulated width: 7.2 ns and measured width: 9.3 ns), respectively. The difference between the measured and simulated pulse widths is mainly reflected in the rising edge. Further, the rise times of the simulated and measured pulses are 1.5 and 4 ns, respectively, and the rising edge of the simulated pulse exhibits transient peak relaxation, which is expected from a laser pulse. However, owing to the circuit test error, the measured pulse does not exhibit this relaxation peak. Additionally, the experimentally detected main-laser pulse is followed by several additional pulses, which are absent in the case of the simulated pulse. These additional pulses in the experimental output can be attributed to interferences caused by the charging and discharging of the capacitor in the actual-driver circuit. Nevertheless, the module successfully emits narrow high-power laser pulses. The good agreement between the simulated and measured data confirms the validity of the laser model as well as the effectiveness of using this model to design high-power driver circuits.

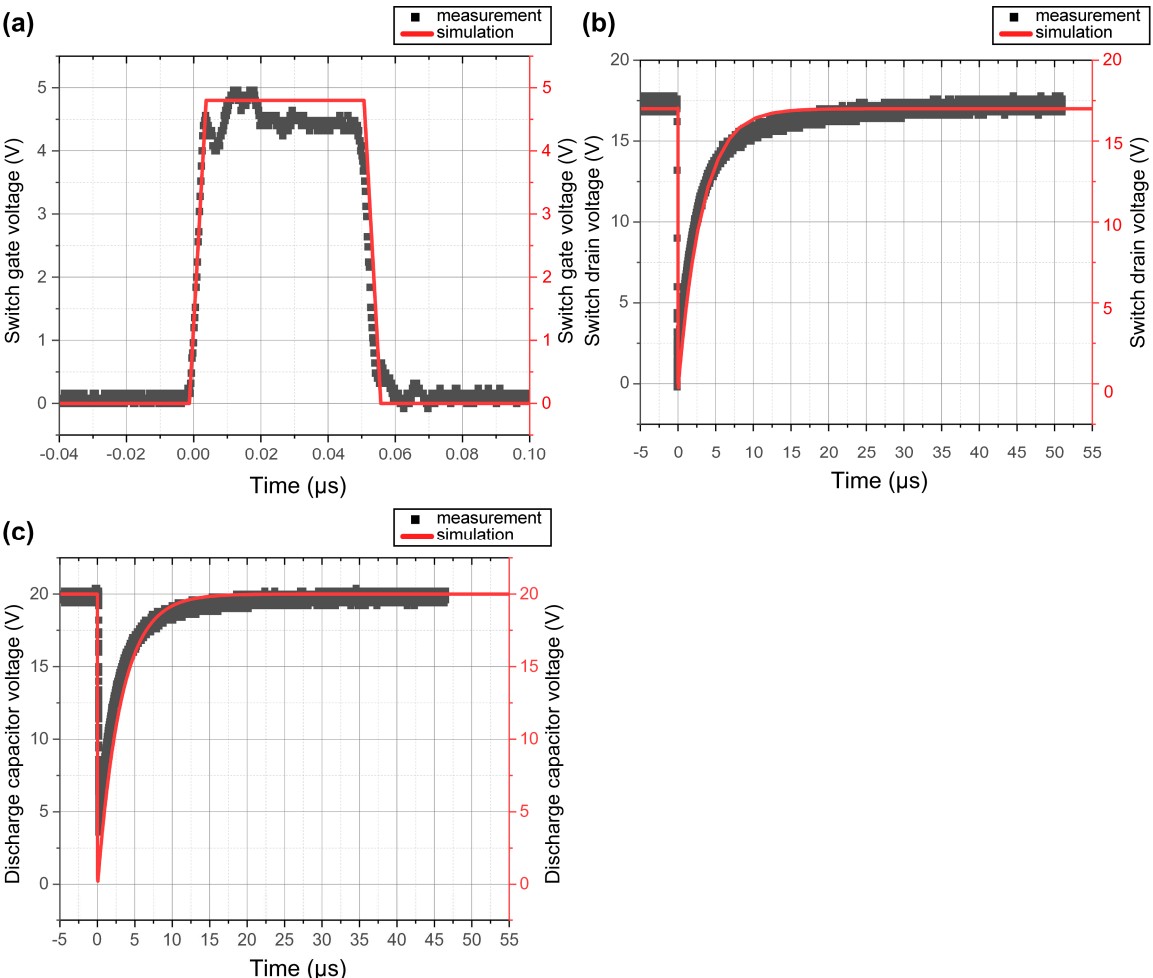

**Figure 14.** Comparison between measured and simulated waveforms of (**a**) switch−gate voltage; (**b**) switch−drain voltage; and (**c**) discharge−capacitor voltage.

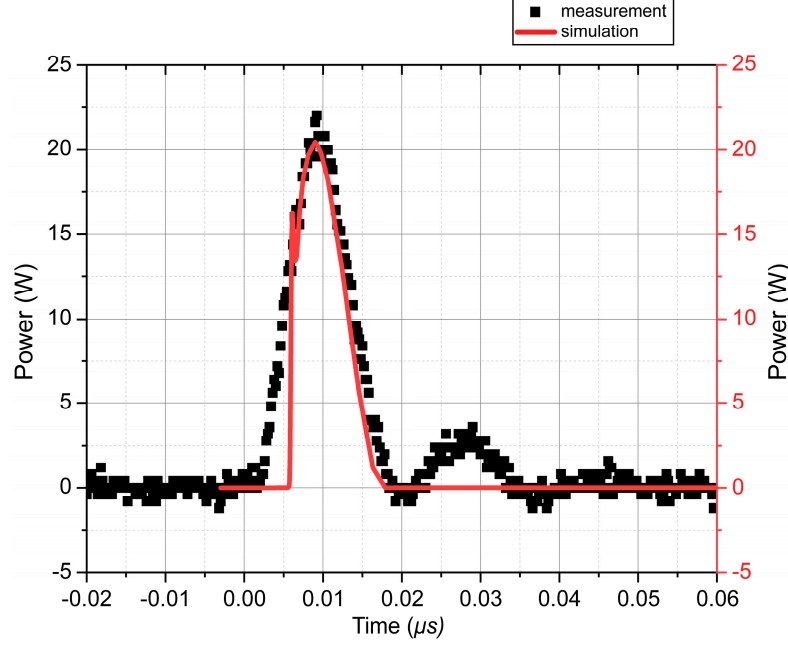

**Figure 15.** Simulated and measured optical-pulse waveforms of the laser−emission module.

## 5. Conclusions

This study presented the design of a 1550 nm pulsed semiconductor laser−emission module based on a multiquantum-well equivalent circuit model. The module design was discussed from two aspects, namely, the laser, and the pulsed drive power. The PI features of the laser model were simulated to optimize the cavity length and the optical facets' reflectivity. The laser model was then introduced into the drive−power design to directly obtain the output waveform characterizing the optical pulse under the pulsed drive, rather than the pulse current that indirectly reflects the optical pulses. Subsequently, the accuracy of the laser model was verified by comparing the simulated PI results of the laser with the PI-feature curves of the laser in actual experiments. Furthermore, critical−voltage and optical-pulse tests were performed on the fabricated physical laser−emission module and were compared to the simulated results. The comparison verified the validity of our design. It is expected that the proposed method can be extended to other structured lasers to facilitate the development of laser−emission modules.

**Author Contributions:** All the authors contributed substantially to this article. The article was conceived and structured by all the authors. Conceptualization, L.L. (Lin Li); methodology, L.L. (Lin Li) and L.L. (Li Li); validation, G.L. and D.W.; formal analysis, L.L. (Li Li); resources, H.X. and L.Y.; data curation, L.L. (Li Li), J.Y. and F.H.; writing—original draft preparation, L.L. (Li Li); writing—review and editing, L.L. (Lin Li). All authors have read and agreed to the published version of the manuscript.

**Funding:** This research received no external funding.

**Data Availability Statement:** Data is unavailable due to privacy or ethical restrictions.

**Conflicts of Interest:** The authors declare no conflict of interest.

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
