# Peer review of "Designing a 1550 nm Pulsed Semiconductor Laser-Emission Module Based on a Multiquantum-Well Equivalent Circuit Model"

_electronics, doi:10.3390/electronics12071578_

Round 1

Reviewer 1 Report

The authors present a work that presents a SPICE model of a MQW laser structure.

The methods are not very clear and derivated by reference 15. The authors used the model proposed in this reference and applied it to an example.

The results are not clear and not reproducible as shown.

The figure 2 is quite confusion as it is not clear what are the different parameters. In particular, the authors define a series of voltage VS1, Vb, ... that are not reported on the diagram. Also, it is not clear how the parameters are obtained.

Therefore, the following simulation is impossible to reproduce. The authors should be clear on how they implemented their SPICE model and what are the parameters they are considering in order for the work to be reproducible.

Figure 3 and Figure 4 are not really helpful as they are well known results from litterature (reference books) and it is not clear how the authors designed their laser out of those graphs?

could they be more explicit on how they have done their design.

No experimental setup is presented and how the experiment has been done. From the datasheet of the components it appears that the maximum current allowed is 18A for the driver in CW and 75A in pulsed mode. Therefore, the measurements have been made in pulsed regime? correct? the authors should clarify their procedure.

No discussion about the results is provided and the authors should discuss their results in relation with the literature. Moreover, it is not clear what are the implications of the measurement results on the design.

Also, the simulations seem under-sampled. Could the authors use a better sampling so the transients can be properly rendered.

Strangely enough, in figure 7, 8 and 9 the laser output exhibit some transient peak relaxation that one expects from a laser and later reported in the measured output.

However, those features disappear in figure 12.

Graphically figure 11 and 12 are bad and should be redone (more points, smoother curves for simulation, better choice of units and ticks)

Figure 13 is redundant and does not provided any new information.

Author Response

Reply to Reviewer 1

  1. The methods are not very clear and derivated by reference 15. The authors used the model proposed in this reference and applied it to an example. The results are not clear and not reproducible as shown.

Reply: Thank you for this insightful comment. As a photonic system, the characteristics of a multi-quantum-well laser can be determined by analyzing a series of rate equations. Considering the integration of a laser and pulse drive circuit, an environment that can simulate electronic and photonic devices and circuits is required. Thus, an equivalent circuit model of multi-quantum-well lasers is established within the framework of the joint solution of the rate and carrier transport equations. To express the transformation process more clearly, we have added a more detailed explanation (Section 2, pages 4–6), reproduced the equivalent circuit diagram (Figure 2), and included the model parameters (Table 1) in the revised manuscript.

  1. The figure 2 is quite confusion as it is not clear what are the different parameters. In particular, the authors define a series of voltage VS1, Vb, ... that are not reported on the diagram. Also, it is not clear how the parameters are obtained. Therefore, the following simulation is impossible to reproduce. The authors should be clear on how they implemented their SPICE model and what are the parameters they are considering in order for the work to be reproducible.

Reply: Thank you for your suggestion. We have redrawn Figure 2 to depict the coupling relationships among the subcircuits. Further, the parameters in the figure correspond strictly to those in the text. The physical meanings and values of the parameters used in the model have been added to the text and are listed in Table 1.

  1. Figure 3 and Figure 4 are not really helpful as they are well known results from litterature (reference books) and it is not clear how the authors designed their laser out of those graphs? could they be more explicit on how they have done their design.

Reply: Thank you for your suggestions. In this study, we used a five-quantum-well chip containing a 7-nm-thick InGaAlAs well layer with a compressive strain of 1.1%, a 10-nm-thick InGaAlAs barrier layer with a tensile strain of 0.6%, an InAlAs–-InGaAlAs graded-index waveguide layer, and an InP cladding. The parameters used in the equivalent circuit model were determined based on the actual chip and are listed in Table 2 in the revised manuscript. Subsequently, the PI features of the established laser model were simulated using SPICE to optimize the cavity length and reflection coefficients. The simulated PI features, with a variable cavity length and reflectivity, exhibited the same trend as that reported in the literature, thus confirming that the equivalent circuit model is also an effective method for device optimization. To further clarify the expression, we have added the reflectance value and cavity length to the captions of Figure 3 and 4 in the revised manuscript, respectively.

  1. No experimental setup is presented and how the experiment has been done. From the datasheet of the components, it appears that the maximum current allowed is 18A for the driver in CW and 75A in pulsed mode. Therefore, the measurements have been made in pulsed regime? correct? the authors should clarify their procedure.

Reply: Thank you for this valuable suggestion. We have included the experimental details in the revised manuscript (section 4, pages 14–16) accordingly.

  1. No discussion about the results is provided and the authors should discuss their results in relation with the literature. Moreover, it is not clear what are the implications of the measurement results on the design.

Reply: Thank you for your suggestion. We have included a detailed discussion on the significance of the obtained results, their relevance in the present study, and their correlation with the results of previously reported studies in the revised manuscript (Section 4, pages 15–17). The difference between the measured and simulated results in Figure 12 indicates that further improvements in the model parameters are required and that the non-thermal saturation mechanism of a semiconductor laser operating at pulsed pump currents should be further explored. The difference between the measured and simulated results in Figure 15 shows that the additional pulses, following the main experimental pulse, are the interference pulses of the circuit, and further optimization of the drive circuit is needed.

  1. Also, the simulations seem under-sampled. Could the authors use a better sampling so the transients can be properly rendered.

Reply: Thank you for your suggestion. We agree with this suggestion, and accordingly, we have modified the simulation results presented in Figures 14 and 15 by increasing the sampling. Further, we have used a line representation for the simulated results to distinguish them from the experimental data (represented by filled squares).

  1. Strangely enough, in figure 7, 8 and 9 the laser output exhibit some transient peak relaxation that one expects from a laser and later reported in the measured output. However, those features disappear in figure 12.

Reply: Thank you for your correction. We examined the corresponding section in Figure 12 and found that after increase the sampling, peak relaxation is visible in Figure 14 in the revised manuscript.

  1. Graphically figure 11 and 12 are bad and should be redone (more points, smoother curves for simulation, better choice of units and ticks). Figure 13 is redundant and does not provided any new information.

Reply: Thank you for your suggestion. We have redrawn Figures 13 and 14 in the revised manuscript (specific changes: increased the sampling of the simulations, changed the Y-axis legend, and removed the time offset). Further, we have removed Figure 13 from the revised manuscript.

Reviewer 2 Report

The authors have proposed an equivalent circuit model of a MQW laser as well as development of a design method for 1550nm pulsed semiconductor laser emission modules.I think this method can potentially improve laser 1550nm laser design which is very important for Telecommunication and sensing in modern technological systems. I strongly believe that the manuscript can provide a useful for laser designers and should be published as it is.

Author Response

Reply to Reviewer 2

The authors have proposed an equivalent circuit model of a MQW laser as well as development of a design method for 1550nm pulsed semiconductor laser emission modules.I think this method can potentially improve laser 1550nm laser design which is very important for Telecommunication and sensing in modern technological systems. I strongly believe that the manuscript can provide a useful for laser designers and should be published as it is.

Reply: Thank you for reviewing our manuscript and summarizing the primary results. We are grateful for your encouraging and positive comments.

Reviewer 3 Report

The generation of high-power ns-duration laser pulses by semiconductor lasers is topical and is being actively developed, especially in the eye-safe spectral range, however, to make the results more clear, a number of additions and comments should be included in the text of the article.
The considered model for simulating pulsed current-voltage curves is one of the possible ones. Optimization of heterostructure parameters, the cavity length and reflection coefficients is carried out within the framework of the joint solution of not only the rate equations and charge carrier transport equations, but also equations that take into account the distribution of photons along the cavity. These studies were carried out for GaAs-based lasers with wavelengths of 850-1060 nm. Therefore, part of the introduction (line 62-65) should be supplemented with references describing simulation models aimed at optimizing heterostructures for high-power pulsed lasers, for example, within the framework of 1D [10.1109/TED.2020.3024353, 10.1109/JQE.2010.2047381] and 2D [doi.org/10.1007/s11082-019-1776-1, doi.org/10.1070/QEL18015] models of semiconductor lasers operating at high pulsed pump currents, when the effects of non-thermal saturation of the watt characteristic appear.
Can the authors provide basic model parameters, including: coefficients used for the gain model, internal optical loss, total optical confinement factor in the MQW of the active region?
Can the authors comment on exactly what effects lead to saturation of the power of the calculated light-current curves?
Can the authors clarify which reflectances were used for each cavity length, when calculating P-I curves shown in Fig. 3?
It is common to call “the cavity surface loss” (line 194) the “output loss”.
Can the authors add the value of the cavity length to the caption of Fig. 4?
Can the authors comment on the difference in the shapes of the theoretical and experimental pulses in Fig. 12, namely, they I am interested in the presence of additional pulses following the main experimental laser pulse and the absence of such pulsations for the calculated laser pulse? Is it possible to present an experimental current pulse that is generated in the pump circuit of a semiconductor laser?
When characterizing ns-laser pulses, the detection technique is also important. Can the authors indicate which photodetectors were used (bandwidth, photodetector connection circuit), as well as the repetition rate of laser pulses and method for experimental determination of the peak optical power and peak pumping current in (fig.10)? It is also useful to give the characteristics of the lasers and heterostructures under study for the CW mode: internal quantum efficiency, internal optical losses, light-current characteristic.

Author Response

Reply to Reviewer 3

  1. The generation of high-power ns-duration laser pulses by semiconductor lasers is topical and is being actively developed, especially in the eye-safe spectral range, however, to make the results more clear, a number of additions and comments should be included in the text of the article.

The considered model for simulating pulsed current-voltage curves is one of the possible ones. Optimization of heterostructure parameters, the cavity length and reflection coefficients is carried out within the framework of the joint solution of not only the rate equations and charge carrier transport equations, but also equations that take into account the distribution of photons along the cavity. These studies were carried out for GaAs-based lasers with wavelengths of 850-1060 nm. Therefore, part of the introduction (line 62-65) should be supplemented with references describing simulation models aimed at optimizing heterostructures for high-power pulsed lasers, for example, within the framework of 1D [10.1109/TED.2020.3024353, 10.1109/JQE.2010.2047381] and 2D [doi.org/10.1007/s11082-019-1776-1, doi.org/10.1070/QEL18015] models of semiconductor lasers operating at high pulsed pump currents, when the effects of non-thermal saturation of the watt characteristic appear.

Reply: Thank you for your helpful suggestions. We have read these suggested papers and found them to be very helpful for improving our paper. Hence, they have been added in the introduction and cited in the main text (Section 4) in the revised manuscript.

  1. Can the authors provide basic model parameters, including: coefficients used for the gain model, internal optical loss, total optical confinement factor in the MQW of the active region? 

Reply: Thank you for your suggestion. The physical interpretation and values of the parameters used in the model have been added to the text and are listed in Table 1 in the revised manuscript.

  1. Can the authors comment on exactly what effects lead to saturation of the power of the calculated light-current curves?

Reply: Thank you for your suggestion. Figure 12 in the revised manuscript compares the measured and simulated power–current curves. Here, the simulated curve shows a certain power saturation, which can be attributed to the gain compression term in the model; this term is widely used as an input parameter in semiconductor laser rate equations. Gain compression is usually determined from the measured small-signal intensity modulation response, and the main mechanism is carrier self-heating. The measured power–current curve did not exhibit saturation, indicating that the time available for heating is insufficient, and thus, the gain compression is negligible. We expect that the difference between the measured and simulated results will promote further studies on improving the model parameters and evaluating the non-thermal saturation mechanism of a semiconductor laser operating at pulsed pump currents.

We have added this information in the revised manuscript (Section 4, page 15).

  1. Can the authors clarify which reflectances were used for each cavity length, when calculating P-I curves shown in Fig. 3?

It is common to call “the cavity surface loss” (line 194) the “output loss”. 
Can the authors add the value of the cavity length to the caption of Fig. 4?

Reply: Thank you for your suggestion. For determining the P-I curves shown in Figure 3 and 4, we used 0.3 as the reflectance value of both the two optical facets and considered a 2-nm-long cavity. These reflectance and cavity length values have been added to the captions of Figure 3 and 4 in the revised manuscript, respectively. Further, we have also replaced “the cavity surface loss” with “output loss” as per your suggestion.

  1. Can the authors comment on the difference in the shapes of the theoretical and experimental pulses in Fig. 12, namely, they I am interested in the presence of additional pulses following the main experimental laser pulse and the absence of such pulsations for the calculated laser pulse? Is it possible to present an experimental current pulse that is generated in the pump circuit of a semiconductor laser?

Reply:

Thank you for your suggestion. We have compared the theoretical and experimental pulses in Figure 15 in the revised manuscript (Section 4, page 17). We considered that the additional pulses, following the main experimental pulse, are the interference pulses caused by the capacitor charging and discharging in the actual driver circuit. In addition, the current of the narrow pulse laser emission circuit was not monitored. However, we obtained the current and optical power pulses from PI test and have represented them in Figure 11.

  1. When characterizing ns-laser pulses, the detection technique is also important. Can the authors indicate which photodetectors were used (bandwidth, photodetector connection circuit), as well as the repetition rate of laser pulses and method for experimental determination of the peak optical power and peak pumping current in (fig.10)? It is also useful to give the characteristics of the lasers and heterostructures under study for the CW mode: internal quantum efficiency, internal optical losses, light-current characteristic.

Reply: Thank you for your suggestion. We have explained the experimental details in the revised manuscript (Section 4, pages 14–16), including the equipment, main parameters (listed in Table 2), connection circuit (Figure 10 and 12), and test method.

We are also grateful for your excellent suggestion to mention the CW lasers characteristics. The following figure shows the power–current characteristics of devices working under the CW and pulse modes. Since the CW characteristics were not used in our study, we have not discussed them in the present manuscript. We also intend to conduct further studies in the near future to calibrate the parameters in the pulse mode.

Reviewer 4 Report

The authors of the manuscript "Designing a 1550 nm pulsed semiconductor laser emission module based on a multi-quantum-well equivalent circuit model" developed an equivalent circuit model for a 1550nm InGaAlAs/InP multi-quantum-well semiconductor laser using rate equations. This model was utilized to optimize the laser drive circuit and reflect the electro-optical features of the laser as an equivalent load. Moreover, the authors investigated the impact of cavity length and facet coating coefficient on the laser output. Following the development of the model, the authors fabricated the laser module and conducted a characterization study. The experimental results closely matched the simulations, demonstrating the accuracy of the proposed model.

The manuscript is well-structured, and the results are effectively presented. The proposed method holds significant potential for researchers in this area, which aligns with the Electronics domain. Thus, I highly recommend publishing this work with some minor modifications that address the comments below:

1. Please adjust the size of Figure 6.

2. Please remove the time offset in Figure 11a and Figure 12. It's hard for readers to get the pulse width from the x-axis. Or put it in the x-label.

Author Response

Reply to Reviewer 4

1.Please adjust the size of Figure 6.

Reply: Thank you for your suggestion. We have adjusted the size of Figure 6 to improve its resolution and visibility.

2.Please remove the time offset in Figure 11a and Figure 12. It's hard for readers to get the pulse width from the x-axis. Or put it in the x-label.

Reply: Thank you for your suggestion. We have accordingly modified Figures 14 and 15 in the revised manuscript (specific changes: removal of the time offset and changing the Y-axis legend).

Round 2

Reviewer 1 Report

The authors improved greatly the manuscript and answered all the comments appropriatly.